# QNCD: Quantization Noise Correction for Diffusion Models

## ABSTRACT

Diffusion models have revolutionized image synthesis, setting new benchmarks in quality and creativity. However, their widespread adoption is hindered by the intensive computation required during the iterative denoising process. Post-training quantization (PTQ) presents a solution to accelerate sampling, aibeit at the expense of sample quality, extremely in low-bit settings. Addressing this, our study introduces a unified Quantization Noise Correction Scheme (QNCD), aimed at minishing quantization noise throughout the sampling process. We identify two primary quantization challenges: **intra** and **inter** quantization noise. Intra quantization noise, mainly exacerbated by embeddings in the resblock module, extends activation quantization ranges, increasing disturbances in each single denosing step. Besides, inter quantization noise stems from cumulative quantization deviations across the entire denoising process, altering data distributions step-by-step. QNCD combats these through embedding-derived feature smoothing for eliminating intra quantization noise and an effective runtime noise estimatiation module for dynamicly filtering inter quantization noise. Extensive experiments demonstrate that our method outperforms previous quantization methods for diffusion models, achieving **lossless** results in W4A8 and W8A8 quantization settings on ImageNet (LDM-4).

## CCS CONCEPTS

• **Computing methodologies** → **Computer vision**.

## KEYWORDS

Diffusion Models, Post Training Quantization, Model Compression

## 1 INTRODUCTION

Recently, diffusion models have achieved remarkable progress in various synthesizing tasks, such as image generating [11], super-resolution [21], image editing and in-panting [27], image translation [23] etc. Compared to traditional SOTA generative adversarial networks (GANs [7]), diffusion models do not suffer from the problem of model collapse and posterior collapse, exhibit higher stability.

However, this comes at the cost of the high computational resources and a large number of parameters required to run these models, which are only available on cloud-based devices. For example, Stable Diffusion [20] requires 16 GB of running memory and

Permission to make digital or hard copies of all or part of this work for personal or classroom use is granted without fee provided that copies are not made or distributed for profit or commercial advantage and that copies bear this notice and the full citation on the first page. Copyrights for components of this work owned by others than the author(s) must be honored. Abstracting with credit is permitted. To copy otherwise, or republish, to post on servers or to redistribute to lists, requires prior specific permission and/or a fee. Request permissions from permissions@acm.org.

*ACM MM, 2024, Melbourne, Australia*

© 2024 Copyright held by the owner/author(s). Publication rights licensed to ACM.
ACM ISBN 978-x-xxxx-xxxx-x/YY/MM
https://doi.org/10.1145/nnnnnnn.nnnnnnn

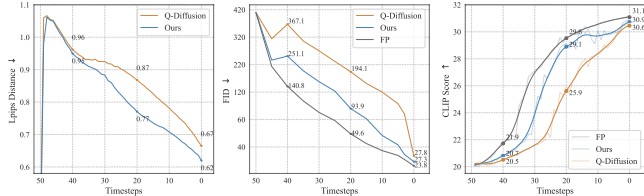

**Figure 1: Comparison of metrics for denoising processes w.r.t. timestep ($t$). LPIPS Distance between the quantized Stable Diffusion model (W8A8) outputs and its floating-point counterpart on MS-COCO, along with their respective CLIP scores and FID (Fréchet Inception Distance) scores.**

GPU and more than 10 GB of VRAM, which is not feasible for most consumer-grade PCs, let alone resource-limited edge devices.

Our work employs post-training quantization (PTQ) to speed up the sampling process in all time steps, while avoiding the high cost of retraining diffusion models. PTQ, having been well-studied in traditional deep learning tasks like classification and segmentation [2, 3, 8], stands out as a preferred compression method due to the minimal requirements on training data and the convenience of direct deployment on hardware devices. Despite the many attractive benefits of PTQ, its implementation in diffusion models remains challenging. The main reason for this is that the framework of diffusion models is quite different from previous PTQ implementations (e.g., CNN and ViT [5, 17] for image recognition). Specifically, diffusion models commonly use UNet structures, which incorporate embedding. In addition, diffusion models iteratively invoke the same UNet model during sampling. In recent work, PTQ4DM [24] and Q-Diffusion [14] first apply PTQ to diffusion models and attribute the challenge to the fact that the activation distribution is constantly changing with time steps. PTQD [9] integrates partial quantization noise into diffusion perturbed noise and proposes a mixed-precision scheme.

In contrast, we analyze in detail the sources of quantization noise and its negative impact on sampling direction, image quality. Specifically, we propose QNCD, a novel post-training quantization noise correction scheme dedicated for diffusion models. First, we identify embeddings in resblock modules as the primary source of intra quantization noise, as embeddings amplifies the outliers of original features, making quantization challenging. We compute smoothing factors from embeddings, making features easy for quantization, thus reducing intra quantization noise. Besides, for inter quantization noise accumulated among sampling steps, we propose a run-time noise estimation module based on the diffusion and denoising theory of diffusion model. By filtering out the estimated quantization noise, our QNCD can dynamically correct deviations in output distribution throughout the sampling steps.

As shown in Fig. 1, with a smaller LPIPS Distance [6] and a higher CLIP Score [19], the sampling direction of our QNCD more closely aligns with that of full-precision (FP) model. In addition, our

method's FID [10] metric consistently outperforms Q-Diffusion, with a final FID reduction of 2.23, reaching 27.33. Overall, our contributions are shown as follows:

- We propose QNCD, a novel post-training quantization scheme for diffusion models to filter out quantization noise.
- We find that a new challenge in quantizing diffusion models is the ongoing emergence and accumulation of quantization noise, which alters sampling direction and final image quality.
- We introduce a feature smooth approach to reduce intra quantization noise when combining features with embeddings. Simultaneously, we utilize a run-time noise estimation module to correct inter quantization noise
- Our extensive experiments show that our method achieves new state-of-the-art performance for post-training quantization of diffusion models, especially in low-bit cases. Additionally, our methodology aligns more closely with the full-precision models in both objective metrics and subjective evaluations.

## 2 RELATED WORK

Model quantization is a method that transitions from floating-point computations to low-precision fixed-point operations. This shift can effectively diminish the model's computational burden, reduce parameter size and memory usage, and expedite computational processes. It can be divided into two main categories: quantization-aware training (QAT) [12] and post-quantization training (PTQ). QAT integrates simulated quantization throughout the training phase. During backpropagation, gradients are calculated to refine the pre-quantized weights, enabling the model to acclimate to quantization errors as training progresses. QAT often yields superior outcomes with significantly fewer bits, but comes with the cost of substantial training overhead and a need for the raw dataset. In constract, PTQ bypasses the need for extensive data retraining, leveraging just a fraction of unlabeled data for calibration, making it a more cost-effective and deployment-friendly alternative. Given that retraining for diffusion has an unaffordable cost (e.g the training of Stable Diffusion [20] requires a cluster of over 4000 NVIDIA A100 GPUs), current works have pivoted towards PTQ to obtain low-bit diffusion models.

Until now, only a handful of current studies have focused on post-training quantization of diffusion models. Among them, PTQ4DM [24] devised a time-step aware sampling strategy for calibration dataset, but its experiments are limited to small datasets and low resolutions. Q-Diffusion [14] employs a state-of-the-art PTQ method (BRECQ [15]) to obtain the performance, which imposes an additional training burden. PTQD [9] integrates partial quantization noise into diffusion perturbed noise and proposes a mixed-precision scheme. TDQ [25] dynamically adjusts the quantization interval based on time step information.

Our method analyze in detail the sources of quantization noise and propose corresponding correction modules. In addition, we employ the most primitive PTQ methods, inherently attributing the performance improvement to our approach.

## 3 METHOD

In Section 3.1, we provide an introduction to the sampling process of the diffusion model and present a unified formula for quantization noise. Following this, in Section 3.2, we analyze the sources of quantization noise and its impact on the sampling direction. In Section 3.3, we present the entire workflow of QNCD.

### 3.1 Preliminaries

Diffusion models are a family of probabilistic generative models that progressively destruct real data by injecting noise, then learn to reverse this process for generation, represented notably by denosing diffusion probabilistic models (DDPMs [11]). DDPM is composed of two chains: a forward chain that perturbs data to noise, and a reverse chain that converts noise back to data. The former is usually designed by hand and its goal is to convert any data distribution into a simple prior distribution (e.g., a standard Gaussian distribution). Given a data distribution $x_0$, the forward process generates a sequence of random variables with transition kernel $q(x_t|x_{t-1})$, as follows:

$$q(x_{1:T}|x_0) = \prod_{t=1}^{T} q(x_t|x_{t-1}),$$
$$q(x_t|x_{t-1}) = \mathcal{N}(x_t; \sqrt{\alpha_t}x_{t-1}, \beta_t I) \tag{1}$$

where $\alpha_t$ and $\beta_t$ are hyper parameters and $\beta_t = 1 - \alpha_t$.

In the denoising process, with a Gaussian noise $x_T$, the diffusion model can generate samples by iterative sampling $x_{t-1}$ from $p_\theta(x_{t-1}|x_t)$ until obtaining $x_0$, where the Gaussian distribution $p_\theta(x_{t-1}|x_t)$ is a simulation of the unavailable real distribution $q(x_{t-1}|x_t)$. The mean value $\mu_\theta(x_{t-1}|x_t)$ of $p_\theta(x_{t-1}|x_t)$ is calculated from the noise prediction network $\epsilon_\theta$ (usually the UNet model):

$$\mu_\theta(x_t, t) = \frac{1}{\sqrt{\alpha_t}}\left(x_t - \frac{\beta_t}{\sqrt{1-\overline{\alpha_t}}}\epsilon_\theta(x_t, t)\right) \tag{2}$$

where $\overline{\alpha_t} = \prod_{i=1}^{t} \alpha_i$. Therefore, the sampling process of $x_{t-1}$ is shown as follows:

$$x_{t-1} = \frac{1}{\sqrt{\alpha_t}}\left(x_t - \frac{\beta_t}{\sqrt{1-\overline{\alpha_t}}}\epsilon_\theta(x_t, t)\right) + \sigma_t z, \quad z \in N(0, I) \tag{3}$$

The diffusion model continuously evokes the noise prediction network to acquire the noise $\epsilon_\theta(x_t, t)$ and filter it out. The huge number of iterative time steps (sometimes $T$=4000) and the complexity of the noise prediction network $\epsilon_\theta$ make the sampling of diffusion models expensive.

Post-training quantization for diffusion models is performed on the noise prediction network $\epsilon_\theta$, which inevitably introduces quantization noise.

$$\widetilde{x}_{t-1} = \frac{1}{\sqrt{\alpha_t}}\left(\widetilde{x}_t - \frac{\beta_t}{\sqrt{1-\overline{\alpha_t}}}\widetilde{\epsilon}_\theta(\widetilde{x}_t, t)\right) + \sigma_t z, \quad z \in N(0, I)$$
$$= \frac{1}{\sqrt{\alpha_t}}\left(\widetilde{x}_t - \frac{\beta_t}{\sqrt{1-\overline{\alpha_t}}}(\epsilon_\theta(\widetilde{x}_t, t) + q_\theta(\widetilde{x}_t, t))\right) + \sigma_t z. \tag{4}$$

The noise prediction network UNet $\epsilon_\theta$ is constructed from multiple Resblocks, where parameterized operations (such as convolutions, fully connected layers, etc.) will generate **intra** quantization

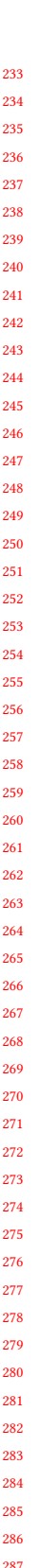

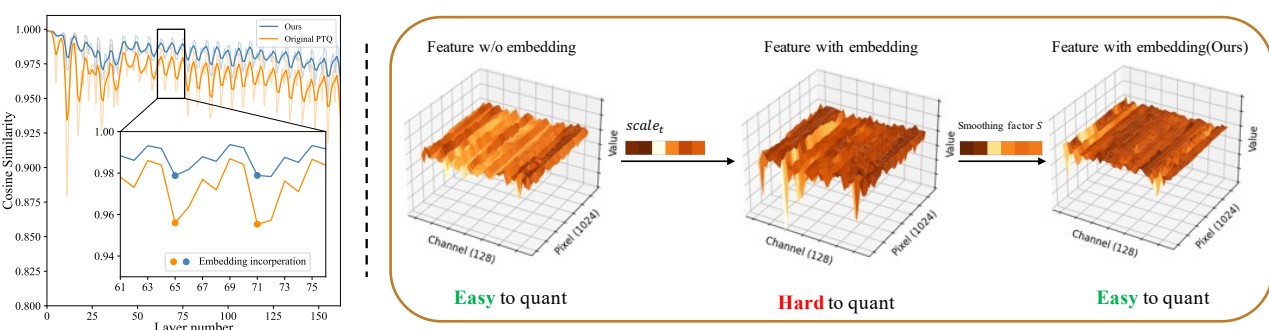

a, Cosine similarity in a single step

b, activation and embedding distribution

Figure 2: (a) shows the similarity of individual layer features across the entire UNet model during a single sampling step, illuminating that quantization noise primarily arises from the incorporation of embeddings. (b) illustrates the distribution of activations before and after the incorperation of embedding (within the last Resblock). When combined with embeddings, outliers in features are amplified, which can be efficiently mitigated using our smoothing factor.

noise within the single-step sampling. From the perspective of complete sampling process, these intra quantization noises accumulate to form **inter** quantization noise $q_\theta(\widetilde{x}_t, t)$ which further accumulates in the current output $\widetilde{x}_{t-1}$, thus affecting subsequent sampling processes.

## 3.2 Quantization noise analysis

*3.2.1 Intra Quantization Noise introduced by embedding.* We consider the quantization noise within the denoising network UNet in a single sampling step as intra quantization noise, which is strongly affected by embedding. As shown in Fig. 2(a), intra quantization noise of the diffusion model exhibits periodic changes during a single step. It escalates when embedding is incorporated into features but then decreases during the fusion of UNet's low-level and high-level features. This cyclical behavior underscores that the primary culprit of quantization noise is the embedding integration phase. Take the embedding fusion stage of DDIM as an example:

$$scale_t, shift_t = layer_{emb}(emb_t).split(),$$
$$h_t = norm(h_t) * (1 + scale_t) + shift_t \quad (5)$$

Where $h_t$ stands for activation and $emb_t$ is the corresponding embedding. The embedding $emb_t$ imparts an utterly different distribution to activation $h_t$. In Fig. 2(b), the distribution of feature $h_t$ generally stabilizes within a quantization-friendly range after processing through a normalization layer. However, the emergence of outliers within the activated correlation channel may pose a challenge.

These outliers are several magnitudes larger than the majority of the data, leading to a skew in the maximum magnitude measurement during quantization. This dominance by outliers could possibly result in reduced precision for the majority of non-outlier values. Further complications arise with the incorporation of embedding, as shown in the middle part of Fig. 2(b). Embedding separates a $1 \times c$ dimensional scale vector $scale_t$ which scales the feature $h_t$ on a channel-by-channel basis. This channel-specific scaling alters the distribution of the activation $h_t$ in a manner that certain channels, especially those with problematic outliers, experience amplification.

Since activations are typically per-tensor quantized, the combined effect of embedding magnification and existing outliers makes the quantization of activations less efficient.

In summary, after normalization, feature $h_t$ is easily quantifiable, but with the introduction of embedding $emb_t$, it becomes challenging to quantify, indicating an increase in intra quantization noise.

*3.2.2 Inter Quantization Noise.* Diffusion models attain their final outputs through iterative denoising network calls. During this procedure, the inter quantization noise, expressed as $q_\theta(\widetilde{x}_t, t)$ in Eq. 4, assimilates into $x_{t-1}$ and advances to the subsequent denoising step, exerting an influence over the entire sampling process.

As shown in Fig. 3, we plot the variation curves of the output Mean and Std during sampling steps. The accumulated inter quantization noise changes the distribution of synthesized data. With continuous sampling, the data distribution of the quantization model further deviates from that of the full-precision model.

## 3.3 Effect of Quantization Noise

Quantization noise reduces the sampling efficiency of the diffusion model, changes the sampling direction, and ultimately reduces the quality of the synthesized image.

First, the introduction of quantization noise gives rise to new noise sources that necessitate denoising, substantially impairing the sampling efficiency of diffusion models. As depicted in Fig. 1, we compare LPIPS distance and FID metrics at each step between the full-precision model and the quantized model. The quantized diffusion model has a FID metric of 194.11 after 30 steps, which is still much larger than the result of the full-precision (FP) model after only 10 steps (FID = 140.76).

Besides, quantization noise also alters the iteration direction of diffusion models. In Fig. 1, we visualize the changing trend of CLIP Score for different methods to provide a more intuitive representation of the iteration direction. At the beginning of the iteration, all methods start with relatively low CLIP Scores, while full-precision

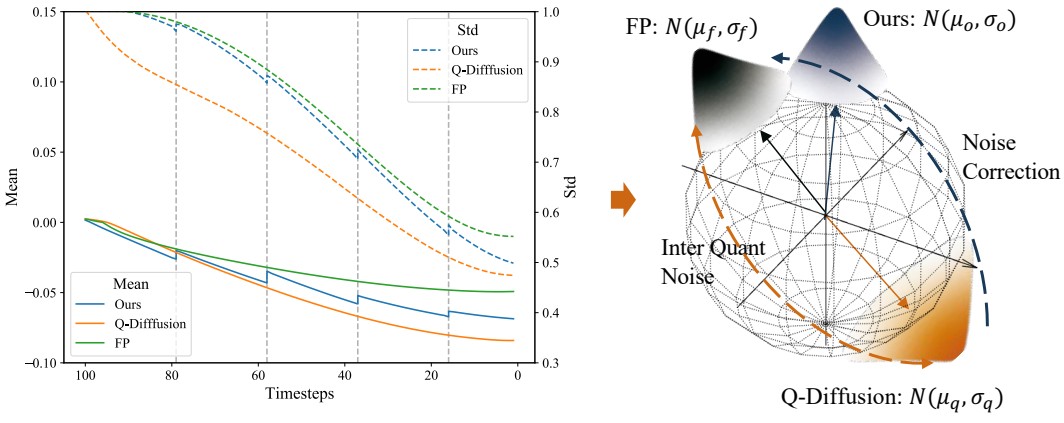

**a, Mean and Std among sampling steps**          **b, Output distribution of a single step**

**Figure 3: (a) demonstrates the mean and std of outputs across all time steps, while (b) visualizes the output distribution at a specific step , revealing a substantial discrepancy between the output of the quantized diffusion model (Orange) and that of the full-precision model (gray). The gray dashed line in (a) represents when our noise estimation module is running.**

model's score continues to rise, indicating correct iteration direction. By step 25, quantized diffusion model shows a 3.16% difference in CLIP Score compared to full-precision model (26.09% vs. 29.25%).

In conclusion, quantization noise presents challenges in maintaining performance following model quantization. This not only calls for minimizing intra quantization noise as much as possible at all steps but also necessitates estimating and filtering out the remaining accumulated inter quantization noise.

## 3.4 Qunantization Noise Correction for Diffusion Models

We propose two techniques: intra quantization correction techniques and inter quantization correction techniques to address the challenges identified in the previous section.

*3.4.1 Intra Quantization Correction.* As shown in Fig.2, during the single-step sampling process, embedding amplifies outliers of activation, leading to an imbalance among channels. Ultimately, this induces a periodic increase in intra quantization noise. For reducing intra quantization noise, we propose the utilization of a channel-specific smoothing factor $S$. By dividing activation with their respective $S$ values, channels are balanced out and more adaptable to quantization. We then incorporate the filtered factor into weights, thus maintaining mathematical equivalence of the convolution, as follows:

$$Y = Q(h_t) * Q(W) = Q(\frac{h_t}{S}) * Q(SW). \tag{6}$$

Ultimately, we can transfer the quantization challenges presented by embedding from activations to weights, which are more robust to quantization.

Since embedding operates on a channel-by-channel basis, our goal is to derive a factor $S$ for each channel from the embedding, making $h_t = h_t/S$ easier to quantize. As evident from Eq. 5, the term $1 + scale_t$, derived from the separation from embedding, accentuates

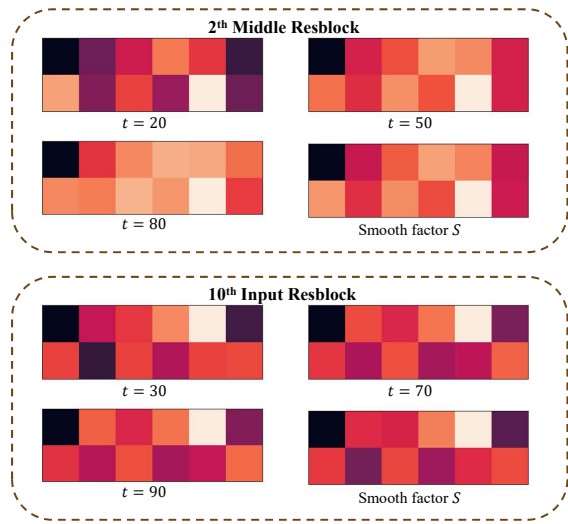

**Figure 4: Visualization of $scale_t$ and smoothing factor $S$ in heatmap representation. For ease of visualization, we select only 12 values from the 512-dimensional data.**

the discrepancies among the activation channels. However, $1 + scale_t$ is dynamic and fluctuates based on the time step $t$. Consequently, we examine the embedding across all $t$ scenarios to ascertain the mean value of $1 + scale_t$ and employ it as a static factor $S$:

$$S = \frac{1}{T} \sum_{t=1}^{T} | 1 + scale_t |, \tag{7}$$
$$scale_t, shift_t = emb_t.split().$$

As shown in Fig. 2(b), The static factor $S$ we obtained serves to calibrate these unbalanced channels, harmonizing the distribution

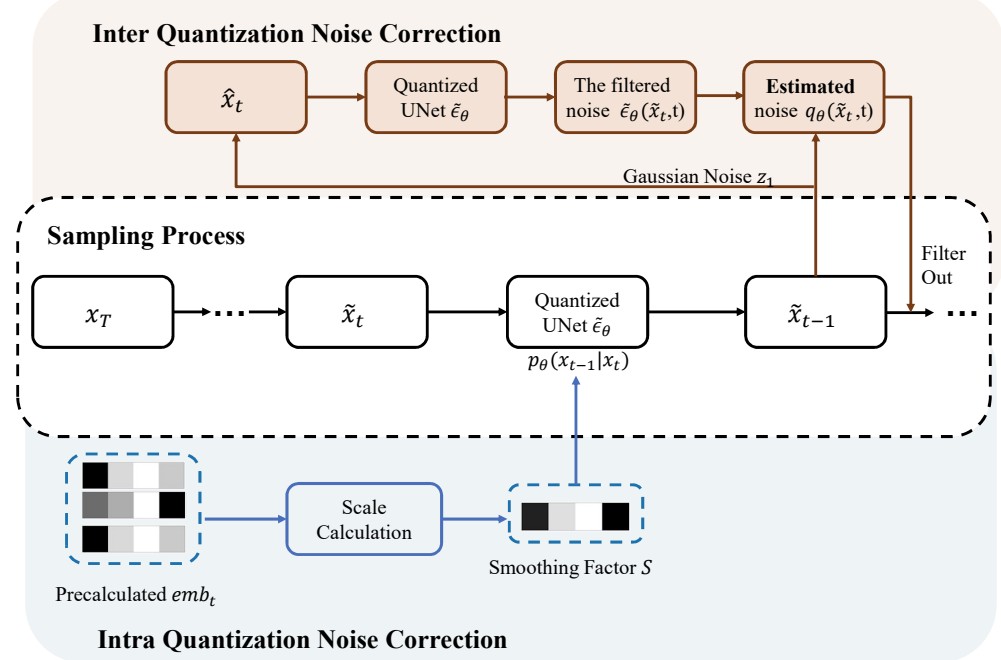

**Figure 5: The pipeline of our proposed method. We initiate by saving the accurate embedding and deduce the smoothing factor $S$ in the calibration stage. During the inference stage, the pre-computed $S$ is applied to smooth the features $h_t$, thereby the intra quantization noise is diminished. Besides, at periodic intervals, the inter quantization noise $q_\theta(\widetilde{x}_t, t)$ is estimated through our noise estimation module, which is filter out in output distribution.**

across each one, rendering the eventual activations more conducive to per-tensor quantization. Different $scale_t$ and $S$ are highly similar, with only minor amplitude differences present on some channels.

In addition, in LDM-type diffusion models, embedding is incorporated differently than in Eq. 5:

$$h_t = \frac{(h_t + emb_t) - \mu_{(h_t + emb_t)}}{\sigma_{(h_t + emb_t)}} * \alpha + \beta, \qquad (8)$$

where $\alpha$ and $\beta$ are pretrained affine transform parameters in the Group-Norm operation. At this point, the distribution of the final activation $h_t$ is jointly determined by $emb_t$ and the coefficients $\alpha$ of group norms. Thus, our smoothing factor $S$ is calculated as follows:

$$S = \frac{1}{T} \sum_{t=1}^{T} \frac{emb_t - \mu_{emb_t}}{\sigma_{emb_t}} * \alpha. \qquad (9)$$

In Fig. 4, we present a visualization of $scale_t$ within $emb_t$ across various modules, time steps, and categories. The key observation is that the distribution of $scale_t$ exhibits imbalance, unevenly scaling the input activations by channel, which renders the activations less suitable for quantization. Despite this, the aggregate distribution of different $scalet_t$ remains relatively stable, with only minor amplitude discrepancies across certain channels. Our smoothing factor $S$ is thus capable of effectively representing the $scalet_t$ encountered during the sampling phase. By filtering through $S$, our QNCD method mitigates the enhancing impact of $salet_t$ on activation outliers, leading to a reduction in intra quantization noise. Crucially, $S$

is pre-computed prior to the formal inference stage, ensuring no additional computational load during the actual inference.

*3.4.2 Inter Quantization Noise Correction.* Eq. 4 shows the process of a single-step sampling in diffusion models, where UNet outputs the filtered noise $\widetilde{\epsilon}_\theta(\widetilde{x}_t, t)$. It consists of two parts, the de-noising noise $\epsilon_\theta(\widetilde{x}_t, t)$, and inter quantization noise $q_\theta(\widetilde{x}_t, t)$, which keeps accumulating in $\widetilde{x}_t$. Ideally, we filter out $q_\theta(\widetilde{x}_t, t)$, thus avoiding the accumulation of quantization noise, but it is impractical to separate $q_\theta(\widetilde{x}_t, t)$ from $\widetilde{\epsilon}_\theta(\widetilde{x}_t, t)$.

The training stage of the diffusion model gives us a possibility to separate $q_\theta(\widetilde{x}_t, t)$:

$$x_t = \sqrt{\alpha_t} x_{t-1} + \sqrt{1 - \alpha_t} z_1 \quad z_1 \in N(0, I). \qquad (10)$$

Eq. 10 does a single-step diffusion operation, where a Gaussian noise $z_1$ is added to $x_{t-1}$ to get $x_t$.

$$L_t^{simple} = E_{t \sim [1,T], x_t, \epsilon_\theta}[||z_1 - \epsilon_\theta(x_t, t)||]. \qquad (11)$$

The training process of the diffusion model (Eq. 11) drives the denoising network $\epsilon_\theta$ to learn the distribution of Gaussian noise $z_1$, which means $\epsilon_\theta(x_t, t) \approx z_1$ is satisfied in the pre-trained diffusion model.

This property is still guaranteed in the quantized pre-trained diffusion model:

$$\hat{x}_t = \sqrt{\alpha_t}\widetilde{x}_{t-1} + \sqrt{1 - \alpha_t} z_1,$$
$$\widetilde{\epsilon}_\theta(\hat{x}_t, t) = \epsilon_\theta(\hat{x}_t, t) + q_\theta(\hat{x}_t, t) \approx z_1 + q_\theta(\hat{x}_t, t). \qquad (12)$$

As shown in Fig. 5 and Eq. 12, we add the standard Gaussian noise $z_1$ to $\widetilde{x}_{t-1}$ to get $\hat{x}_t$, and feed it into the quantized model $\widetilde{\epsilon}_\theta$. The output of the quantized model contains both the Gaussian noise $z_1$ to be filtered out and the newly introduced quantization noise $q_\theta(\hat{x}_t, t)$. $\hat{x}_t$ is obtained by a single-step denoising and diffusion process on $\widetilde{x}_t$, thus their distributions remain highly similar as well as the corresponding quantization noise:

$$q_\theta(\widetilde{x}_t, t) \approx q_\theta(\hat{x}_t, t) \approx \widetilde{\epsilon}_\theta(\hat{x}_t, t) - z_1. \tag{13}$$

Finally, the quantization noise $q_\theta(\widetilde{x}_t, t)$ can be determined, as the Gaussian noise $z_1$ is manually designed and $\widetilde{\epsilon}_\theta(\hat{x}_t, t)$ is the output of the noise predicting network, both of which are ascertainable.

Based on the above analysis, we can estimate the quantization noise by simulating the diffusion model training process. We first add the deterministic noise $z_1$ to $\widetilde{x}_{t-1}$, which can be filtered out in the diffusion sample process, and then the rest of the indeterministic noise is the quantization noise $q_\theta(\widetilde{x}_t, t)$. Besides, the quantization noise is obtained through estimation and doesn't align perfectly with the actual noise in terms of pixel dimension, whereas it is identical at the level of the overall distribution. Thus we get the distribution of the quantization noise at stage $t-1$ and correct the output sample in Eq. 4.

The above procedure is for single-step quantization noise estimation. In the diffusion model, the distributions of samples from neighboring steps are very similar, as well as the corresponding quantization noise distribution $(q_\theta(\widetilde{x}_{t+1}, t+1) \approx q_\theta(\widetilde{x}_t, t))$. Therefore, in actual sampling process, we divide entire sampling steps into multiple stages, estimating the distribution of quantization noise one time per stage. For example our method run only 4 times to estimate the quantization noise during a sampling process of 100 time steps, which brings a negligible increase in sampling duration. As shown in Fig. 3, our QNCD periodically estimates the distribution of inter quantization noise and corrects distribution deviations caused by this noise. This noise correction enables the distribution of sample outputs to closely align with the full-precision models.

## 3.5 Summary of methods

As shown in Fig. 5, our method contains two major blocks: intra quantization noise correction module and inter quantization noise correction module. Firstly, we determine the smoothing factor $S$ on a channel-by-channel basis, consequently transitioning the distribution disparities induced by embedding over to the weights, which makes the activation easier for quantization. Secondly, we discern the distribution of quantization noise via our run-time noise estimation module, enabling its exclusion in subsequent sampling steps.

## 4 EXPERIMENTS

## 4.1 Implementation Details

**Datasets and quantization settings:** Consistent with the experimental details of PTQ4DM [24], Q-Diffusion [14], we conduct image synthesis experiments using pre-trained diffusion models (DDIM [26]), latent diffusion models (LDM [20]) and Stable Diffusion on four standard benchmarks: CIFAR(32×32) [13], ImageNet (256×256) [4], LSUN-Bedrooms(256×256) [29], MS-COCO

(512×512) [16]. All experimental configurations, including the number of steps, variance, etc., follow the official implementation. To facilitate quantification and comparison of the validity of the methods, we use the most naive PTQ method (mse-based range setting) in the 8-bit case, which is simple and fast. For the case where the weights are quantized to 4bit, we adopt BRECQ [15] as well as Adaround [18] to ensure quantization model performance, in consistency with Q-Diffusion. In addition, we sample uniformly from all time steps to obtain the calibration dataset for PTQ with 5120 samples on all datasets.

**Evaluation Details:** Consistent with PTQ4DM and Q-Diffusion, for each experiment we report the widely adopted Frechet Inception Distance (FID) [10] and sFID [22] to evaluate performance. For ImageNet and CIFAR experiments, we additionally report Inception Score (IS) [1] for reference to ensure consistency of reported results. For MS-COCO, we introduce CLIP Score to ensure the correspondence between the synthesized images and prompts.

In line with Q-Diffusion, we generated 50,000 samples for evaluating our method. However, the sampling process for diffusion models is time-consuming, especially for high-resolution images such as MS-COCO(512×512). Consequently, in the experiment where Stable Diffusion is used for generating MS-COCO, we produce only 10,000 samples to speed up comparative process.

## 4.2 Unconditional Generation

**Results on CIFAR:** The results are displayed in Tab 1. Note that W$n$A$m$ means $n$-bit quantization for weights and $m$-bit quantization for activations. It can be seen that at the W8A8 bitwidth, our method achieves FIDs and sFIDs that are very close to the full-precision model, with FID reductions of 0.57 (steps=100) and 0.38 (steps=250) compared to Q-Diffusion. In addition, previous methods confronted great difficulties in mitigating large amounts of quantization noise due to low-bit quantization. For example, in the settings of W4A6 and 100 steps, Q-Diffuison obtains FIDs and sFIDs as high as 39.07 and 43.36, implying that the large amount of activation noise leads to a performance breakdown. While our method conducts a detailed analysis of quantization noise and effectively eliminates it, it still achieves a lower FID value of **12.26**, proving the effectiveness of our method. Our method performs 6-bit quantization of activation on diffusion models and ensures that the performance does not collapse, whereas previous methods have been performed at 8-bit.
**Results on LSUN-Bedrooms:** At the W8A8 bitwidth, our method reduces the FID by 0.21 compared to Q-Diffusion as shown in Tab. 2, proving the effectiveness of our method on the task of high-resolution image synthesis.

## 4.3 Class-conditional Generation

**Results on ImageNet:** we carry out complex experiments on the generation of conditional ImageNet datasets to demonstrate the effectiveness of our method. To facilitate the validation, we adopt the LDM-4 model with 20 steps. As shown in the Tab. 2, our method consistently narrows the performance gap between quantized and full-precision diffusion models. Specifically, under the settings of W8A8 and W4A8, our QNCD can approach **lossless** performance. It is worth noting that at the W4A6 bidwidth setting, the IS of Q-Diffusion drops to 89.82, which is 156.1 lower than that of the

**Table 1: Quantization results on CIFAR(32 × 32) with DDIM. (50,000 samples)**

| Method | Bitwidth (W/A) | DDIM(Steps=100) | | | DDIM(Steps=250) | | |
|---|---|---|---|---|---|---|---|
| | | IS ↑ | FID ↓ | sFID ↓ | IS ↑ | FID ↓ | sFID ↓ |
| FP | 32/32 | 9.04 | 4.19 | 4.41 | 9.06 | 4.00 | 4.35 |
| TDQ | 8/8 | 8.85 | 5.99 | - | - | - | - |
| Q-Diffusion | 8/8 | 9.17 | 3.93 | 4.34 | 9.38 | 3.84 | 4.27 |
| Ours | 8/8 | **9.24** | **3.36** | **4.24** | **9.41** | **3.46** | **4.21** |
| Q-Diffusion | 4/8 | 9.41 | 4.92 | 5.13 | 9.64 | **4.37** | 4.59 |
| Ours | 4/8 | **9.53** | **4.85** | **5.06** | **9.78** | 4.43 | **4.51** |
| Q-Diffusion | 4/6 | 7.53 | 39.07 | 43.36 | 7.81 | 34.65 | 37.29 |
| Ours | 4/6 | **8.86** | **12.26** | **14.83** | **9.01** | **11.09** | **13.46** |

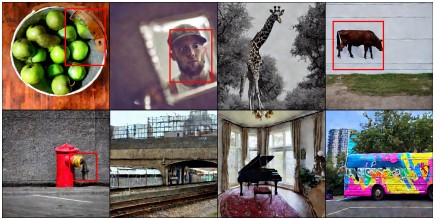

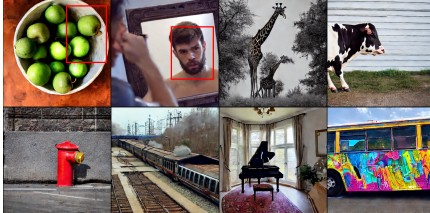

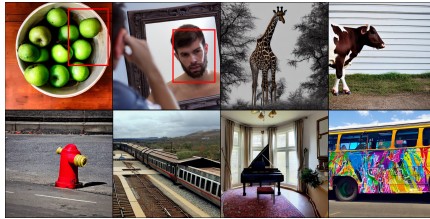

(a) Q-Diffusion

(b) Ours

(c) Full Precision

**Figure 6: Stable Diffusion 512 × 512 text-guided image synthesis results using Q-Diffusion and our QNCD under W8A8 precision. All text prompts are sourced exclusively from MS-COCO dataset.**

**Table 2: Comparisons with extra SOTA methods on ImageNet (LDM-4,Steps=20) and LSUN-Bed (LDM-4,Steps=200). "*" means results in the corresponding paper.(50,000 samples)**

| Method | ImageNet(FID ↓ / IS ↑) (FP:11.42/245.39) | | | LSUN-Bed(FID ↓ / SFID ↓) (FP:3.16/7.84) |
|---|---|---|---|---|
| | W8A8 | W4A8 | W4A6 | W8A8 |
| PTQD* | 11.94/153.92 | 10.40/214.73 | - | 3.75/9.89 |
| TDQ* | - | - | 41.23/- | - |
| Q-Diffusion | 10.92/229.31 | 9.56/219.64 | 41.25/89.82 | 4.03/10.15 |
| QNCD | 10.57/231.85 | 9.48/221.62 | 20.14/136.49 | 3.82/**9.65** |

full-precision model (245.39). Our method well handles the low-bit quantization of the activation. Compared to the FID of Q-Diffusion which is as high as 41.25, the FID of our method is **20.14**, indicating the effectiveness of our method. The visualizations are available in the Appendix.

## 4.4 Text-guided Image Generation

We assess the performance of QNCD through Stable Diffusion for text-guided image generation, using text prompts derived from the MS-COCO dataset. As demonstrated in Tab. 7, our method surpasses Q-Diffusion in both FID metrics and CLIP Scores.

In addition, we visualize the final generated image in Fig. 6. For all three methods (FP, Q-Diffusion, and ours), we have given the same content conditions as input to facilitate comparison. It can be noticed that the accumulated quantization noise changes the content space of the image, causing the final synthesized image to be shifted. As shown in the red box in Fig.6, the synthesized image shows the importation of abnormal content , such as abnormal faces,

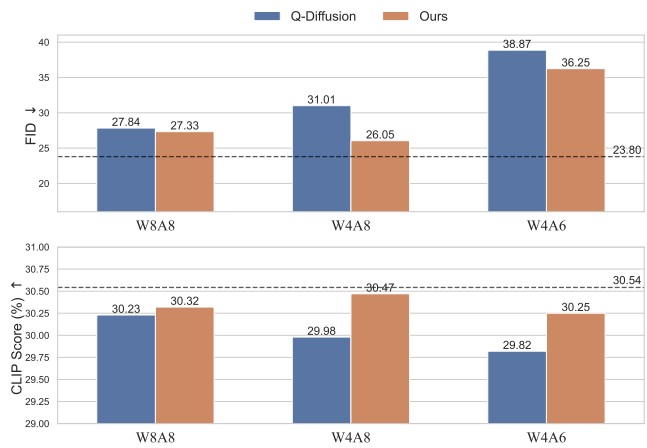

**Figure 7: Quantization results for Stable Diffusion(steps=50) on MS-COCO (10,000 samples). The dashed lines represent results of full-precision model.**

incomplete bowls and floating cows. Compared with Q-Diffusion, our method provides a higher quality image, which is closer to the full-precision model synthesized image and has more realistic details, colors, and richer semantic information. In conclusion, our method effectively mitigates the quantization noise, and is closer to the full-precision model not only in terms of **statistical metrics**,

**Table 3: The effect of different modules of QNCD with Stable Diffusion on MS-COCO(512×512).**

| Method | Bitwidth | Stable Diffusion(Steps=50) | |
| --- | --- | --- | --- |
| | (W/A) | FID ↓ | CLIP Score ↑ |
| FP | 32/32 | 23.80 | 30.54 |
| Q-Diffusion | 8/8 | 27.84 | 30.23 |
| Intra-QNCD | 8/8 | 27.41 | 30.25 |
| Inter-QNCD | 8/8 | 27.60 | 30.29 |
| QNCD | 8/8 | **27.33** | **30.32** |

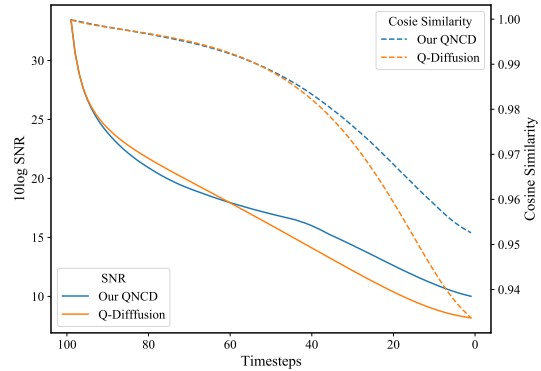

**Figure 8: Comparison of the signal-to-noise- ratio (SNR) and cosine similarity in each step of DDIM(100) on CIFAR.(W8A8)**

but also in terms of **visualization**. More visualization results are shown in Appendix.

## 4.5 Ablation Study

*4.5.1 Comparison of SNR and Cosine Similarity.* In Fig. 1, we present a visualization of the LPIPS Distance between the quantized model output and its floating-point counterpart for all 100 time steps, demonstrating that our method consistently yields more accurate results.

Given the output of the floating-point noise prediction network and the corresponding quantization noise, we plot the change curve of the signal-to-noise ratio (SNR) of the quantized diffusion model, as shown in Fig. 8, as well as the cosine similarity between the corresponding output and the FP output. We can find that: 1, with the increase of denoising steps, the cosine similarity between the quantized model output and the FP output is continuously decreasing, which also means that the overall quantization noise is continuously accumulating, and the corresponding SNR is continuously decreasing. 2, our method can estimate and filter out the quantization noise on a global scale, resulting in a better SNR, and a sampling process closer to that of the floating-point model.

*4.5.2 Effects of each module.* As shown in Tab. 3, we perform ablation experiments on Stable Diffusion (step=50) of MS-COCO 512×512 dataset to demonstrate the effectiveness of our proposed method. Our QNCD method consists of two parts, intra quantization noise correction (Intra-QNCD) and inter quantization noise

**Table 4: Inference performance and Image Quality Assessment(IQA) for MS-COCO via Stable Diffusion (512∗512, 50 steps).**

| | FP16 | Original PTQ(W8A8) | Q-Diffusion(W8A8) | QNCD(W8A8) |
| --- | --- | --- | --- | --- |
| Inference Time | 959.5ms | **601.8ms** | 628.3ms | 631.2ms |
| IQA Score↑(0 ~ 1) | 0.847 | 0.728 | 0.775 | **0.793** |

correction (Inter-QNCD). By using Intra-QNCD, we achieve a reduction of 0.43 in FID compared to Q-Diffusion. And our Inter-QNCD is able to reduce 0.24 in FID and improve 0.06 in CLIP Score. By introducing both blocks, our method QNCD achieves a reduction of 0.51 in FID, showing that these two blocks can collaborate to achieve higher performance improvement. These results demonstrate the effectiveness of our proposed techniques for noise correction in post-training quantization of diffusion models.

*4.5.3 Comparison of real inference efficiency.* For fair comparison, we provide end-to-end inference times in Tab. 4. Inference times are based on the UNet of Stable Diffusion V1.4, which denote whole denoising process of diffusion models. The experimental background is A100, TensorRT-8.6 and CUDA-11.7. Similarly to our QNCD, Q-Diffusion introduces the Short-Cut split operation in pursuit of better model performance, which also imposes an additional inference burden (26.5ms compared to original PTQ). Our method runs at a similar speed to Q-Diffusion, but with higher image quality.

*4.5.4 Comparison through Image Quality Assessment.* As shown in Tab. 2, the FID metrics of PTQD and Q-Diffusion on ImageNet dataset are 11.94 and 10.92, superior to the FP's score of 12.45 under the W8A8 setting. This same pattern extends to the LSUN-Bedrooms and CIFAR datasets, which is **unexpected** and implies that the FID metric may not be an optimal indicator of image quality. This is because FID focuses more on the overall distribution similarity rather than the specific quality of each image. For a more comprehensive comparison, we further refer to objective **Image Quality Assessment**(IQA) metrics proposed in CLIP-IQA [28] to evaluate 5000 synthesized images. Our method achieves an IQA metric of 0.793, which is better than Q-Diffusion (0.775), but still falls short compared to FP (0.847).

## 5 CONCLUSION

In this paper, we propose QNCD, a unified quantization noise correction scheme for diffusion models. To start with, we do a detailed analysis of the sources and effects of quantization noise in terms of visualization and actual metrics, and find that the periodic increase in intra quantization noise comes from embedding's alteration of feature distributions. Thus, we calculate a smoothing factor for features to reduce quantization noise. Besides, a run-time noise estimation module is proposed to estimate the distribution of inter quantization noise, which is further filtered out in the sampling process of the diffusion model. Leveraging these techniques, our QNCD surpasses existing state-of-the-art post-training quantized diffusion models, especially at low-bit activation quantization (W4A6). Our approach achieves the current SOTA on multiple diffusion modeling frameworks (DDIM , LDM and Stable Diffusion) and multiple datasets, demonstrating the broad applicability of QNCD.

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
