# OpenReview forum: "QNCD: Quantization Noise Correction for Diffusion Models"
_acmmm.org/ACMMM/2024/Conference — MM2024 Poster_

### Official Review · Reviewer_aU1n · 2024-05-21

**Rating:** 5
**Confidence:** 3

**Summary:**

The authors propose a quantization noise correction method (QNCD) for post-training quantization (PTQ) of diffusion models. The authors identify two primary quantization challenges: intra and inter quantization noise. QNCD combats these two types of noise through embedding-derived feature smoothing for eliminating intra quantization noise and an effective runtime noise estimation module for dynamically filtering inter quantization noise.

**Strengths:**

1) The QNCD is a novel post-training quantization scheme for diffusion models to filter out quantization noise.
2) The authors introduce a feature smooth approach to reduce intra quantization noise when combining features with embeddings. Besides, the authors utilize a run-time noise estimation module to correct inter quantization noise.
3) The experimental results seem good.

**Limitations:**

Basically, the manuscript is interesting. I have some comments:
1) What is the period of QNCD for correcting inter quantization noise during experiments? In Sec. 3.4, the authors adopt 4 times during a sample process of 100-time steps. With more QNCD steps, will the performance keep growing?
2) The authors use 10000~50000 samples for image generation in experiments. However, in Fig. 3 and 8, the authors only conduct 100 sampling steps. Will the phenomenon within 100 sampling steps still be obvious when sample steps are large enough?
3) There are some typos, for example:

In abstract, “noise estimatiation” -> “noise estimation”.

In abstract, “dynamicly filtering” -> “dynamically filtering”.

**Suitability:**

3

---

### Official Review · Reviewer_bgUj · 2024-05-24

**Rating:** 4
**Confidence:** 3

**Summary:**

The paper introduces a novel Quantization Noise Correction Scheme (QNCD) to address challenges in post-training quantization (PTQ) for diffusion models. Motivated by the need to reduce computational intensity while maintaining sample quality, the study identifies and tackles intra and inter quantization noise issues. The key contributions include proposing QNCD, identifying the emergence and accumulation of quantization noise in diffusion models, introducing techniques to reduce intra quantization noise, and demonstrating state-of-the-art performance in low-bit quantization settings. Overall, the paper significantly advances post-training quantization methods for diffusion models, offering promising solutions for accelerating sampling without sacrificing quality.

**Strengths:**

1. The paper exhibits exceptional readability, effectively conveying the background, challenges, and methodology even to those less familiar with quantization-related work. The data analysis presented is well-structured and persuasive, providing valuable assistance to readers in understanding the paper.
2. The two proposed solutions, Intra Quantization Correction and Inter Quantization Noise Correction, appear highly practical. Their simplicity of operation and minimal overhead make them urgently needed in the industry.
3. The idea of Inter Quantization Noise Correction is indeed bold and insightful! The use of approximate symbols reflects the authors' deep understanding of quantization noise in diffusion models. This method has a wild beauty that doesn't care about strict formula derivation.
4. The experimental results are impressively remarkable.

This work holds significant reference value for the industry. If the authors can address my concerns to further substantiate its practicality, I am inclined to recommend that this submission should definitely be presented at ACM Multimedia 2024. (To be honest, I don't think all of these concerns can be answered.)

**Limitations:**

1. For the Intra Quantization Correction, I notice that the specific implementation is $Y=Q(\frac{h_t}{S})*Q(SW)$, as shown in Eq. (6). While the quantization burden of the first term $\frac{h_t}{S}$ is indeed reduced, I wonder if this might transfer the quantization burden to $SW$?

2. I observe that the static factor $S$ for diffusion models seems to be calibrated based on statistical results. It might be necessary to explain why the static factor $S$ can always be calibrated. **What's more, please explain the advantage of Intra Quantization Correction compared to per-channel quantization.** These questions could potentially impact the practicality and novelty of this method.

3. Although I appreciate the idea behind the Inter Quantization Noise Correction, it seems more like an intuitive leap and lacks evidence to support its accuracy. The explanation of Inter Quantization Noise Correction in the supplementary material remains inadequate, especially in the presence of typos in Eq. (2).

4. I am not entirely certain whether this work considers the quantization noise $q_{\theta}(\hat{x_t},t)$ as a variable or a distribution. **I assume that the Inter Quantization Noise Correction must have done something right to lead to the observed performance improvement. However, I find it difficult to comprehend how subtracting the quantization noise can filter $q_{\theta}(\hat{x_t},t)$ out when $q_{\theta}(\tilde{x_t},t)$ "doesn't align perfectly with the actual noise in terms of pixel dimension".** In my understanding, $q_{\theta}(\hat{x_t},t)$ and $q_{\theta}(\tilde{x_t},t)$ (or $q_{\theta}(\tilde{x_{t-1}},t-1)$ ) have the same (or similar) distribution but are independent of each other (since $z_1$ and $z$ are independent). In mathematics, the result of subtracting two independent random variables follows the distribution that is the convolution of their individual distributions, which seems to indicate that the subtraction method described in this paper (Eq. (4) in the *Supp.*) is wrong. I hope that the details and principles of Inter Quantization Noise Correction can be explained more clearly, which will probably help improve my evaluation of this paper.

**Suitability:**

3

---

### Official Review · Reviewer_zpQc · 2024-05-25

**Rating:** 4
**Confidence:** 2

**Summary:**

The study introduces the Quantization Noise Correction Scheme (QNCD), a post-training quantization strategy for diffusion models aimed at addressing the challenges posed by quantization noise during the image synthesis process. Diffusion models are known for their high-quality image synthesis capabilities but suffer from extensive computational demands, particularly in the iterative denoising stages. QNCD specifically targets the reduction of both intra and inter quantization noise, which typically degrade the quality of synthesized images. Intra quantization noise, exacerbated by embeddings in the resblock module, and inter quantization noise, arising from cumulative quantization errors across the denoising process, is mitigated through embedding-derived feature smoothing and a dynamic runtime noise estimation module, respectively. Extensive experiments confirm that QNCD not only outperforms existing quantization methods but also achieves lossless results in low-bit settings, effectively aligning with full-precision models in both objective metrics and subjective evaluations across various datasets and diffusion modeling frameworks.

**Strengths:**

1. QNCD introduces a groundbreaking solution to the significant challenge of quantization noise in diffusion models.
2. The scheme effectively addresses both intra and inter quantization noise through feature smoothing and dynamic noise estimation.

**Limitations:**

1. The dual-component approach of QNCD, while effective, may introduce complexities in implementation, particularly in integrating feature smoothing and noise estimation into existing diffusion model architectures. There are a few details in Figure 5. It would be better to provide more details about the implementation.

2. It would be beneficial to explore whether QNCD can be applied to image restoration tasks. Given its effectiveness in enhancing image quality in low-light enhancement settings by addressing quantization noise, applying QNCD to image restoration could potentially yield similar improvements. Exploring this application could broaden the utility of QNCD and potentially lead to significant advancements in the field of image restoration.

**Suitability:**

2

---

### Official Review · Reviewer_1VBQ · 2024-05-27

**Rating:** 5
**Confidence:** 4

**Summary:**

This paper introduces a novel post-training quantization scheme specifically designed for diffusion models called Quantization Noise Correction Scheme (QNCD). This paper identifies a critical challenge in the quantization of diffusion models: the continuous emergence and accumulation of quantization noise that distorts the sampling direction and degrades image quality. To address these issues, the paper presents a feature smoothing approach to mitigate intra-quantization noise during feature embedding, and a run-time noise estimation module to dynamically correct inter-quantization noise. The extensive experiments conducted demonstrate that QNCD achieves state-of-the-art performance in post-training quantization for diffusion models and closely aligns with full-precision models in both objective metrics and subjective evaluations.

**Strengths:**

1.	The manuscript exhibits excellent readability, allowing readers to easily grasp the intentions of the work. The analysis of the challenges in quantizing diffusion models is clear, and the data analysis is convincing and well-substantiated.
2.	The two proposed solutions, Intra Quantization Correction and Inter Quantization Noise Correction, appear simple yet are practical and effective. The Inter Quantization Noise Correction is particularly interesting.
3.	The experimental results presented in the paper are impressively effective.

**Limitations:**

1.	The paper appears to be overly engineering-focused, particularly in the solution for Intra Quantization Correction. Different diffusion model weights exhibit varying characteristics, and a static factor S may not be universally applicable. It raises the question of whether there may be diffusion models for which a reasonable static factor S does not exist.
2.	Despite being supported by experimental results, the methods proposed in the paper still seem to lack theoretical substantiation. The use of three `$\approx$` in Eq. (13) is particularly confusing. Considering that quantization errors are not inherently large, it is unconvincing to disregard the discrepancies between variables masked by the use of `$\approx$`.
3.	The y-axis (FID) of the middle subplot in Figure 1 is non-linear, and the numerical intervals do not seem to match the y-axis (27.8, 27.3, 23.8). It is recommended to check for potential numerical errors.

**Suitability:**

3

---

### Meta-Review · Area_Chair_aapU · 2024-06-29

**Recommendation:** Accept (Poster)
**Confidence:** 5

**Metareview:**

The paper received unanimous acceptance from all reviewers. The AC agrees but notes that the methods being compared are somewhat outdated. Therefore, AC encourages the authors to cite and include comparisons with the latest state-of-the-art methods, such as those published in ICLR.